# Assessing Managed Aquifer Recharge Processes under Three Physical Model Concepts

**Thomas Fichtner \*, Felix Barquero, Jana Sallwey and Catalin Stefan**

Research Group INOWAS, Department of Hydrosciences, Technische Universität Dresden, 01069 Dresden, Germany; felix.barquero@mailbox.tu-dresden.de (F.B.); jana.sallwey@tu-dresden.de (J.S.); catalin.stefan@tu-dresden.de (C.S.)

\* Correspondence: thomas.fichtner@tu-dresden.de; Tel.: +49-351-463-44168

**Abstract:** Physical models such as surface infiltration experiments in the lab and field are an approach to understand processes in the unsaturated soil zone. In the case of mapping processes influencing the operation of real-world managed aquifer recharge schemes they are helpful tools to determine interactions between processes in the unsaturated soil zone, and site-specific as well as operational parameters. However, the multitude of assumptions and scale-related limitations of downscale investigations often lead to over- or underestimations, rendering their results useless when translated to field-like conditions. Various real-world managed aquifer recharge operational scenarios were simulated in three physical models, a 1D-lab column, a rectangular shaped stainless steel 3D-lab infiltration tank and a rectangular shaped 3D-field unit, to understand the impact of the experimental set-up on the assessment of processes and to identify the experimental set-up which is most-suitable to describe these processes. Results indicate that water flow velocity, water saturation and oxygen consumption are often overestimated in 1D-column experiments due to sidewall effects and no existing lateral flow. For precise analysis of infiltration processes in general as well as during operation of managed aquifer recharge, 3D experiments are recommended due to their more realistic representation of flow processes.

**Keywords:** managed aquifer recharge; infiltration experiments; physical models; experimental set-up; flow processes; oxygen consumption

## 1. Introduction

Excess withdrawal of groundwater from aquifers leads to declining groundwater levels worldwide as more water is being consumed than can be renewed by nature [1]. Especially in Asia, Arab countries and North and Central America, overexploitation is caused by water abstraction for irrigation or for direct industrial water supply [2–6]. Moreover, the advancing climate change, along with changing precipitation patterns influence the replenishment of groundwater resources negatively [7–10]. A side effect of declining groundwater levels is the deterioration of groundwater quality due to, for example, saltwater intrusion [5,6,11–13].

Managed aquifer recharge (MAR) is a measure to reverse or mitigate negative effects on groundwater resources caused by overexploitation and climate change [14–16]. It implies the use of excess surface water to recharge an aquifer under controlled conditions for later use or environmental benefit [17]. MAR often involves large-scale facilities, the most common techniques being injection wells, infiltration ponds and galleries or recharge dams [18].

In the case of using infiltration basins, two factors are of prime importance during the operation—the infiltration quantity and the quality improvement of the infiltrated water during the soil passage. MAR is one of the measures that can be implemented to secure water supply,

compensate for some effects of climate change and, more generally, handle the quantity and quality of groundwater bodies. Nevertheless, some MAR technologies can also be used to limit the pollution of surface water by infiltrating some of the polluted water and monitoring the geo-purification and/or attenuation processes. Therefore, MAR can also be undertaken to protect the environment by limiting the level of pollution in sensitive receptor media. These include, for example, groundwater contamination by recharge water that contains pollutants (trace metals, metalloids, microorganisms, pharmaceutical products, etc.). Both factors are influenced by clogging processes which are caused by existing suspended solids and gas bubbles in the infiltrated water, by precipitation of minerals as well as by the growth of bacteria in the soil matrix [19]. The development of physical, mechanical, chemical and biological clogging leads to changes in the soil pore system, thus resulting in the reduction of infiltration capacity and the decrease of transport of water and air into the soil matrix. Clogging was found to be the most common reason for the shutdown of MAR projects [20] followed by economic or political reasons [21]. Thus, minimization of clogging is one of the essential prerequisites of any project. This is a complex task, however, as clogging rates depend on a combination of different site-specific conditions such as soil properties, climate, water quality and process-related parameters such as hydraulic loading rate (HLR), the annual infiltrated amount of water, and hydraulic loading cycle (HLC), which relates to the ratio of infiltration phase and following dry phase.

To assess and ideally control clogging processes for a MAR site, models are used to reproduce these phenomena. For the experimental design, different temporal and spatial scales are being utilized. Often, laboratory experiments are conducted before undertaking tests on the actual MAR site. Field tests are time-consuming, costly and it is usually impractical to assess the processes in detail in the natural environmental conditions where they occur. Most models are built to simulate the effect of a change of the influential conditions on the system, in order to predict the effects of these changes on the real system. Another advantage of a model is that phenomena can be studied under controlled conditions, e.g., different climate scenarios, which can be merely impossible to reproduce in the field. Results from physical models include information on the reduction of infiltration capacities due to clogging processes as well as on the purification capacities of the soil during MAR, which is influenced by different site specific and operational parameters.

Understanding these processes not only helps to adapt the design of a MAR site but also indicates the maintenance and operational costs [22]. For the assessment of the main processes taking place during operation of MAR, mostly laboratory [23,24] and pilot scale experiments [25,26] are carried out to characterize these processes under different boundary conditions [27–29]. For the characterization of more specific processes, such as clay dispersion or metals release, often further small-scale batch and column experiments are performed [30–35]. Clay dispersion, affected by the presence of water with higher salt concentration [30,33], can lead to soil pore blockage as well as consequent reductions in soil saturated hydraulic conductivity whereas mobilization of metals from sediments such as arsenic [32,36–38] and also iron and manganese [39] poses a challenge to maintaining local groundwater quality.

However, it is known that a multitude of assumptions and scale-related limitations of small-scale investigations lead to over-or underestimations of processes taking place in soil during infiltration of water [40] and it was stated that the extrapolation of controlled laboratory investigations to the field scale is highly uninvestigated [27]. Only a few authors [41,42] discussed the limitations of transferring the results from laboratory clogging experiments to the field.

The most critical design issues referring to unsaturated soil laboratory experiments are the existence of unnatural preferential flow paths caused by sidewall flow, scaling issues due to spatial variations and the presence or absence of preferential flow [43]. Sidewall flow is not representative of the full-scale field conditions and leads to a preferential flow of fluid close to the outer wall of a soil column or lysimeter due to prevented horizontal water flow leading to lower residence time in the soil [43–45].

Regarding column experiments, it was observed that clogging processes tend to be skewed towards the source of water [40,46]. Biological clogging will be biased towards the inflow, accumulating biological substrate in the inlet area of the soil column. Microbiological growth further along the column will be underestimated. Long-term column studies showed, however, that over the course of time, clogging will propagate further into the column [29]. Even though column studies cannot reproduce all of the main trends of clogging observed during field trials, they show great potential for the estimation of key clogging parameters, such as degradation rates of particles and growth rate of biofilm [46].

Considerations of seasonal climate conditions at the MAR site are important as they influence biological clogging processes. Different water temperatures can significantly affect the bacterial growth and metabolism behavior and further induce differences in the biological clogging rate and process [47]. Higher temperatures in summer lead to faster and stronger biological clogging than in winter, because of a faster cultivable bacteria growth rate and extracellular polymeric substance (EPS) production. Moreover, the strength of biological clogging depends also on changing direct sun exposure caused by different day length and radiation power. It was demonstrated by [48] that clogging processes in winter and summer are governed by different processes. Furthermore, temperature of infiltrated water can influence water viscosity, which has an impact on soil hydraulic permeability [49]. It was observed that colder temperatures in winter lead to a decrease of hydraulic conductivity in infiltration basins by a factor of 1.5 to 2 in comparison with summer [50]. Additionally, surface water temperature variation over the day may cause changes in water infiltration rates [49]. On the contrary, other studies [51,52] demonstrated that temperature changes are not the dominating force for changes of hydraulic conductivity.

Comparison of clogging processes in laboratory and field studies is not straightforward due to the often different climate conditions. Less direct sun exposure and temperature changes in the laboratory compared to the field can lead to underestimation of the clogging strength in laboratory experiments. Furthermore, seasonal changes of temperature cause changes in water viscosity what may lead to over- or underestimation of clogging effects. These aforementioned issues indicate that extrapolation of results from laboratory scale MAR experiments to larger-scale field systems must be evaluated carefully. Limitations und uncertainties related to scale and experimental setup are not fully understood, and guidelines on advantages and disadvantages of the different systems are needed.

To understand the restrictions and possibilities of different MAR experiments, vadose zone experiments in different scales (field, laboratory), dimensions (1D, 3D) and under the influence of different climates (cold: <10 °C, mild: 10–17 °C, warm: >18 °C) were set up. Results were compared concerning the reproduction of soil clogging processes, water flow, oxygen dynamics and degradation of organic substances during operation. This study highlights the unique setup of the three physical models reproducing MAR basin infiltration experiments as well as displays first results from the experiments conducted. Indications for further utilization of the experimental setup and the suitability assessment of different laboratory and field experiments for simulating processes taking place in the unsaturated soil zone during MAR operation are given.

## 2. Materials and Methods

### 2.1. Experimental Setup

Three experimental units with different dimensions were constructed to address the study aims (Figure 1):

- a cylindrical plastic 1D-column in the laboratory (Length (L) 1 m, Diameter (D) 0.15 m),
- a rectangular-shaped, stainless steel 3D-infiltration tank in the laboratory (Length (L) 1.5 × Width (W) 1.0 × Hight (H) 1.0 m) and
- a rectangular-shaped 3D-infiltration unit in the field without confinement (L 4.5 × W 3.0 × H 1.0 m)

The experimental setup of the field infiltration unit was chosen to allow a direct comparison of field processes with downscaled laboratory conditions. Except for the height of the infiltration unit, all dimensions were changed by a factor of three. Since representation of field processes in a tank is a rather uncommon, the experimental setup was complemented by a column with the same height as the tank. This study aims to assess whether reducing the processes from a 3D to a 1D setup are justified and what restrictions are produced by this simplification.

The column and tank were placed inside of a fully automatic climate tent, which facilitates the control of air temperature and humidity, whereby the field scale unit was influenced by the local climate temperature, humidity and radiation (temperate continental climate). The influence of precipitation events was prevented by covering the basin with a transparent mobile roof.

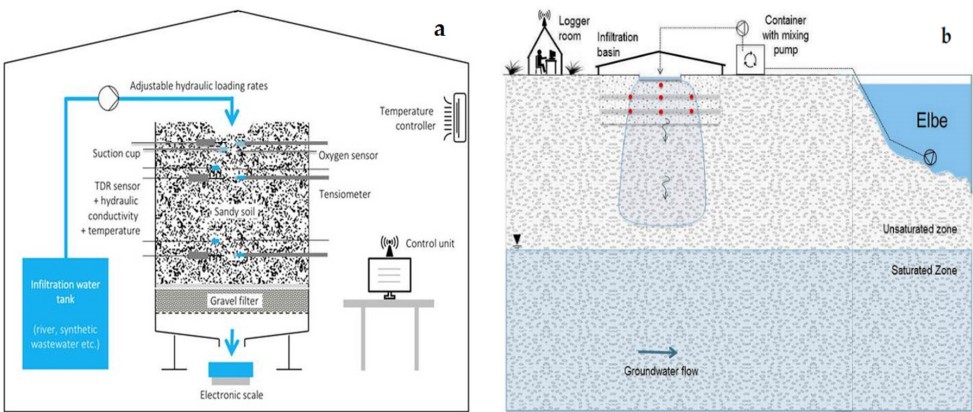

**Figure 1.** Experimental set-up of infiltration laboratory scale unit (**a**) and infiltration field scale unit (**b**).

### 2.1.1. Installation of the Soil Material

All experimental units were packed with the same sandy soil (K-value: $6 \times 10^{-5}$ m/s; 91.2% sand, 8% silt, 0.8% clay), with a filling height of 85 cm. The soil material was placed on a filter disc (column) and on a grating with gravel filter (tank). The soil material in the field infiltration unit was placed directly on the underlying natural soil material. In conformity with the natural compaction of similar soil types (according to [53]), the placement of the soil was undertaken with the objective of acquiring a bulk density of 1.6 g/cm$^3$. After completion of each scenario, the soil material of the upper layer was removed and refilled to ensure the absence of soil material affected by clogging processes remained in the experimental units.

### 2.1.2. Infiltration Area

After completion of the packing, the infiltration basins were constructed (Figure 2). In the case of the laboratory tank and field infiltration unit, the infiltration was regulated through recharge basins installed in the center of their surface (dimensions of laboratory tank: L 0.45 × W 0.30 × H 0.06 m$^3$, in field infiltration unit: L 1.35 × W 0.90 × H 0.06 m$^3$). The infiltration in the lab column took place by flooding the soil surface, a common approach in column experiments [27,47].

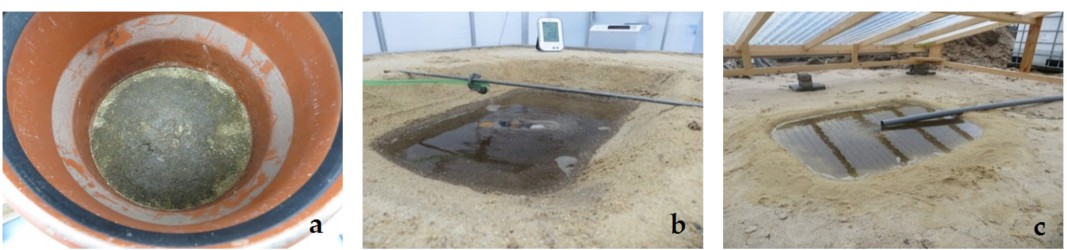

**Figure 2.** Infiltration basin: column (**a**), laboratory tank (**b**) and field infiltration unit (**c**).

### 2.1.3. Measurement Devices

For the estimation assessment of spatial and temporal distribution of soil moisture characteristics, tensiometers (tensio160, UGT GmbH, Müncheberg, Germany) and TDR-probes (UMP-1, UGT GmbH, electrical conductivity and temperature measurement included) were installed in the three experimental units. Additional fiber-oxygen sensors (OXROB3, Pyroscience GmbH, Aachen, Germany) were installed in the column and tank to measure oxygen availability. The recording of data was undertaken every 2 min by data loggers (DL-200, UGT GmbH and FireStingO2, Pyroscience GmbH). Suction cups for taking soil water samples were included to control degradation of infiltrated substances.

### 2.1.4. Placement of Sensors

Besides the geometry and dimension of the experimental units, the location and number of measurement devices were planned carefully to define their optimal placement and to avoid measurement errors. Thus, prior to installation of the sensors, simulations of soil moisture characteristics were undertaken with the software Hydrus 2D/3D [54]. 26 possible sensor locations were arranged in three layers below ground level (0.16, 0.28 and 0.68 m) to compare the pressure head dynamics for different infiltration scenarios. Based on the simulation results [55], which indicate a remarkable influence on the pressure heads measurements if the sensors are situated closed to the border (<10 cm), one set of tensiometer and a TDR-probe were installed in the centre of the infiltration basin and two of each sensors were installed at the lateral sides. Thus, at least 0.25 m distance to the wall of the infiltration unit was kept (Figure 3). Simulations showed that the arrangement of three sensors (TDR-probe and tensiometer) per horizontal layer is sufficient to detect the changes in water content during infiltration by basins [55]. In the case of the column, only one sensor per layer was installed due to the reduced available space.

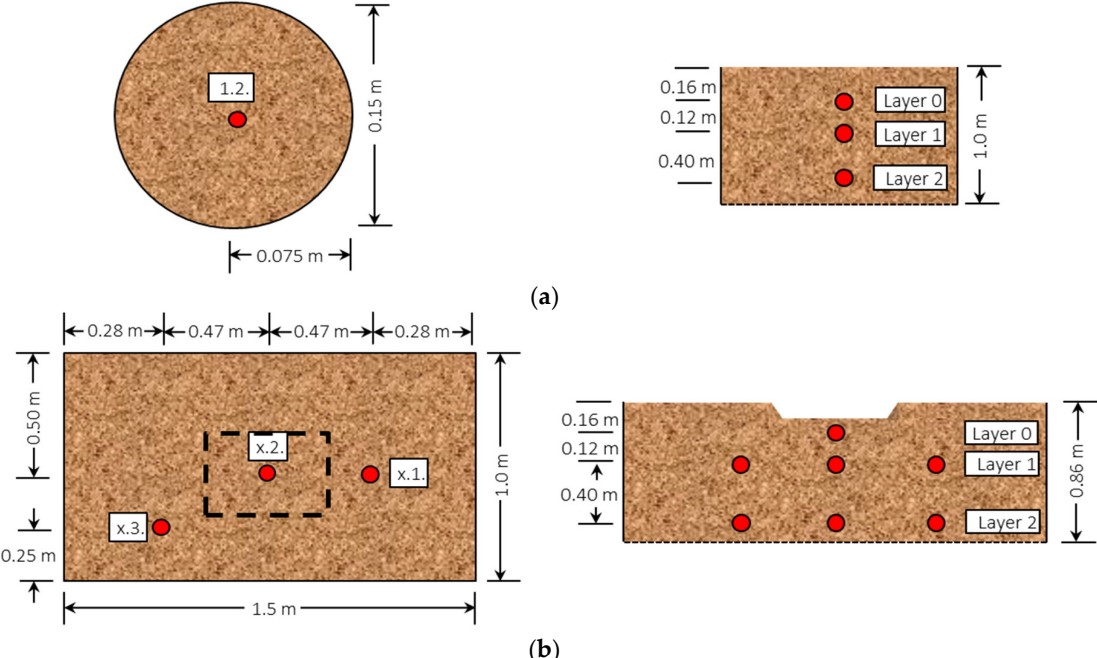

**Figure 3.** *Cont.*

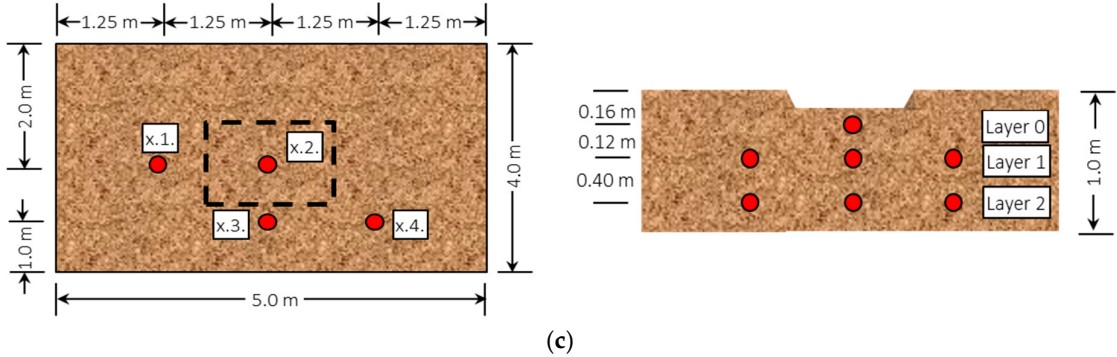

**(c)**

**Figure 3.** Sketch for the placement of sensors in column (**a**), tank (**b**) and field infiltration unit (**c**).

Oxygen probes and suction cups were installed at the same positions in three different depths of 0.16, 0.28 m and 0.68 m to identify possible correlations between soil moisture, oxygen availability as well as soil water quality.

### 2.2. Operation System

River water (Elbe River) with an average dissolved organic carbon (DOC) concentration of 7.5 mg/L and a total suspended solids concentration of 5 to 15 mg/L was infiltrated with an hydraulic loading rate of 300 m/a, chosen based on preliminary infiltration tests. The wet/dry ratios, the ratio between length of infiltration phase and following dry phase, were changed between the scenarios (Table 1), to show the interscaling effect of different factors: infiltration capacity reduction, water flow, and oxygen consumption.

**Table 1.** Boundary conditions in the experimental units during the infiltration scenarios.

| Scenario | Wet/Dry Ratio = HLC (-) | Climate | | | |
|---|---|---|---|---|---|
| | | Lab Scale Infiltration Units | Field Scale Infiltration Unit | | |
| 1 | 1:3 (24 h/72 h) | | 1 (cold) | 2 (mild) | 3 (warm) |
| 2 | 1:1 (168 h/168 h) | 2 (mild) | Temp. < 10 °C | Temp. 10 to 17 °C | Temp. > 18 °C |
| 3 | 1:- (648 h:0 h) | Temp. 17 °C | Humidity 85% | Humidity 83% | Humidity 79% |
| 4 | 3:1 (72 h/24 h) | Humidity 70% | Solar irradiance < 34 W/h | Solar irradiance 34 to 230 W/h | Solar irradiance > 230 W/h |

The infiltration in laboratory column and tank took place at a constant temperature (17 °C) and humidity (70%) which complies with defined climate 2, whereas the scenarios in the field were performed under three different seasonal climates with changing median temperature, humidity and solar irradiance (Table 1). All scenarios were stopped when the infiltration units were overflowing or if the change in reduction of infiltration capacity between two consecutive infiltration cycles was smaller than 10 percent.

### 2.3. Determination of Infiltration Capacity Reduction

There is a variety of techniques available for determining variable hydraulic capacity in the unsaturated soil zone such as single/double infiltration ring, tension disk infiltrometer or constant/falling head well permeameter methods [56–60]. However, these techniques influence the water content in the unsaturated soil zone, the soil compaction as well as the physical state of the clogging layer, which could affect the results of the investigations negatively. An alternative for determining the clogging rate or the percentage reduction of the infiltration capacity is the performance of tracer tests [61,62].

For this study, easily traceable NaCl (concentration 1 g/L) was added to the infiltrated water for determining the changing median flow velocity. The median runtime of the tracer ($t_{50}$) was monitored

by changing electrical conductivity through a number of tracer experiments per scenario. The median flow velocity of the infiltrated water was calculated for different observation points.

## 3. Results and Discussion

### 3.1. Infiltration Capacity Reduction

The results (Figure 4) show that there is a visible reduction of infiltration capacity by 20 to 90% in all three experimental units due to clogging processes. However, clogging processes in exemplarily represented scenario 1 and 2 differ in extent of reduction but also in delay of infiltration capacity reduction.

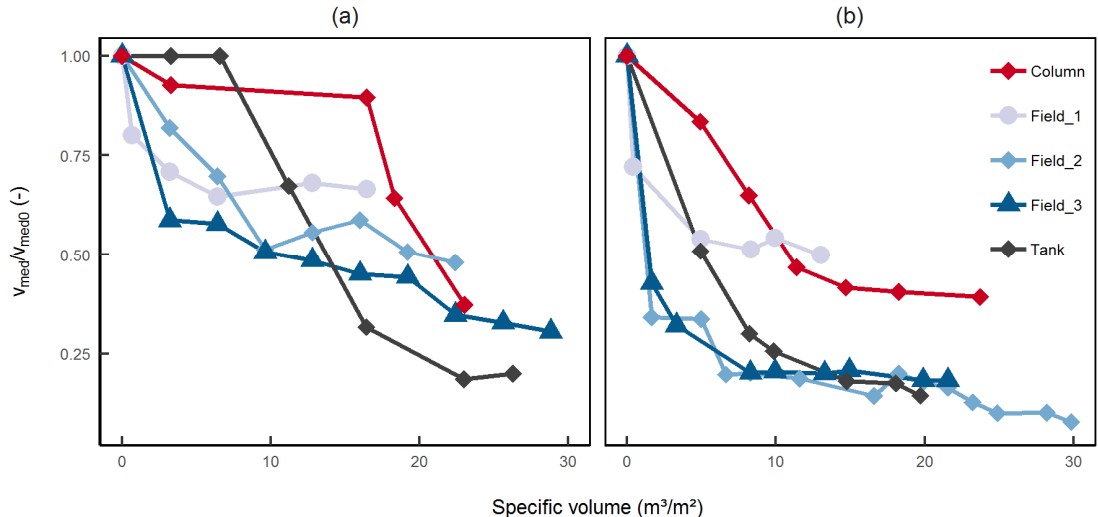

**Figure 4.** Relative reduction of median flow velocity in point 1.2 in (**a**) scenario 1 (wet/dry ratio: 24 h/72 h) and (**b**) scenario 2 (wet/dry ratio: 168 h/168 h) for the column and tank under climate 2 (mild) as well as the field under climate 1 (cold), 2 (mild) and 3 (warm).

Scenario 1 was undertaken with 1 day infiltration followed by 3 days drying phase. The results show that the reduction of infiltration capacity is less significant than in scenario 2 and that the tank as well as the column have a delay in the infiltration capacity reduction in comparison to the field. In the tank reduction started after infiltration of 7 $m^3/m^2$; in the column, significant reduction only occurred after 17 $m^3/m^2$. The observed delay in infiltration capacity reduction in comparison to the field is attributed to different clogging processes governing the reduction in the various setups. Due to the reduced direct sunlight in the laboratory, which is necessary for the growth of clogging causing algae, physical clogging processes will govern the reduction in the tank and the column. Biological clogging resulting from algae growth starts after a longer lag phase at the beginning of the infiltration (Figure 4a). In the field, clogging processes will be caused by a combination of physical and biological clogging, which leads to an immediate steeper decrease of infiltration capacity. The combination of two clogging processes contributes to the instant infiltration capacity loss whereas physical clogging in the laboratory evolves more slowly.

Scenario 2 was undertaken with infiltration for 7 days, followed by 7 days of the drying phase. Reduction of infiltration capacity could be observed right from the beginning of the infiltration and it dropped to 50% after the infiltration of 5 $m^3/m^2$, with the exception of the column experiment. Results of the column experiment indicate a delayed and lower reduction (50% reduction) in median flow velocity. The shapes of the curves indicate that a combination of physical and biological clogging is responsible for the reduction of hydraulic conductivity from the beginning of infiltration. Furthermore, it can be seen that the decrease of infiltration capacity in column and tank is lower compared to the field due to less biological clogging as demonstrated by [28]. This could be caused by restricted growth

conditions for algae in the laboratory experiments. Sensitivity of clogging processes to changing climate can be recognized by comparison of the different graphs for field experiments (Figure 4, Field_1, Field_2, and Field_3). The reduction in the field under colder climate (Field_1) presented the lowest reduction in infiltration capacity of all experimental setups. The colder climate combines lower temperatures with less solar irradiance and thus leads to a reduced growth of clogging caused by algae. Here, the effect of temperature changes on the infiltration capacity seems to be dominated by the reduced biological clogging and not by the competing effects of increased water viscosity.

Furthermore, the results show that the reduction of infiltration capacity under climate 2 and 3 in the field was very similar. From these observations, it can be concluded that an increase of temperature and sun irradiance does not lead automatically to further reduction of the infiltration capacity. On the one hand, the influence of temperature and sun irradiance on biological clogging may be limited by threshold values for temperature and sun irradiance and an exceedance would lead to no more reduction of infiltration capacity. On the other hand, the influence of higher temperatures on biological clogging may be overlain by temperature effects on the water viscosity and consequently on the water flow velocity. Thus, the similarity of hydraulic conductivity reduction under the warm and mild climate might be caused by a combination of clogging effects and decreasing water viscosity as reported by [39,40].

## 3.2. Water Flow

The hydraulic behavior of all experimental setups was assessed to understand the different flow regimes in the experimental setups and parameters affecting them such as the sidewall effects. The arrival times of the wetting front at point 2.2 were determined for the column, tank and field experiment based on matric potential measurements (Figure 5).

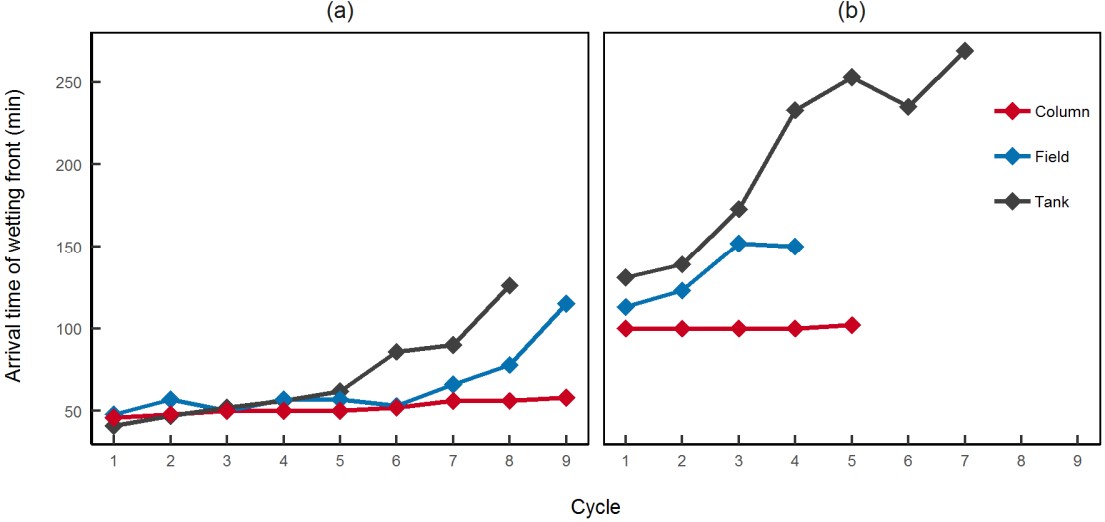

**Figure 5.** Increase of flow time between soil surface and point 2.2 for (**a**) scenario 1 (wet/dry ratio: 24 h/72 h) and (**b**) scenario 4 (wet/dry ratio: 72 h/24 h). Scenarios were aborted when overflow was detected, leading to different numbers of cycles for the setups.

This first noticeable difference in scenario 1 and 4 is that the arrival time of the wetting front for the latter is more than two times (>100 min) above the value of scenario 1 (50 min). This can be attributed to the hydraulic loading rate staying constant for both scenarios (300 m/a) and the simultaneous infiltration duration changing from 1 day (scenario 1) to 3 days (scenario 4). Three times lower infiltration rates lead to 2–3 times greater arrival times for the wetting front.

Despite observed clogging in all the three experimental units, combined with ponding of water during the infiltrations, the arrival times of the wetting front at point 2.2 did not increase in the column as in the tank and the field experiment (Figure 5a). In scenario 1, the arrival time for tank and field

increased by 100 min whereas in the column it stays relatively constant. In scenario 4, the arrival time in the tank increased by 150 min, in the field by 50 min and in the column again it stays constant. For scenario 4, both the field and the column scenario had to be stopped due to overflow. The column results show a significant difference as despite detected clogging, the water movement in the column stays constantly fast. Preferential flow paths, such as along the side walls, could contribute to this phenomenon. Thus, the applicability of column experiments for the determination of flow durations is limited and must be evaluated carefully. While the tank and field show similar tendencies in increasing flow arrival times over the course of MAR experiments, they differ in the intensity and point in time when the increase begins. General mechanisms are therefore represented by both systems but the different boundary conditions cause a divergence.

General differences in hydraulic behavior can also be identified by comparison of matric potential measurements schemes at point 2.2 (Figure 6). The general reaction of all setups to drying and wetting phases is clearly depicted. The matric potential follows the scheme of rapid decrease after the infiltration start, a phase of relatively constant low values during infiltration, followed by a phase of rapid increase and finally a slower increase phase during drying. This scheme can be reproduced with all experimental setups. Differences lie within the maximum and minimum matric potential values. Column values always lie beneath those of the field and tank measurements. The strongly limited lateral flow in the column leads to higher water contents in the wet and dry phases of the scenarios compared to the field and tank where the lateral water flow is not restrained. Although matric potential values in the infiltration phase are relatively similar for the field and tank measurements, they differ strongly during the drying phase. The field drains stronger than the tank, leading to values that are 80% higher than those measured in the tank despite underlying soil layers with similar hydraulic characteristics. However, drainage in the tank is based on a grating with gravel filter, which could lead to the built-up of a capillary barrier and thus unexpected high saturation values in the lower soil layers of the tank.

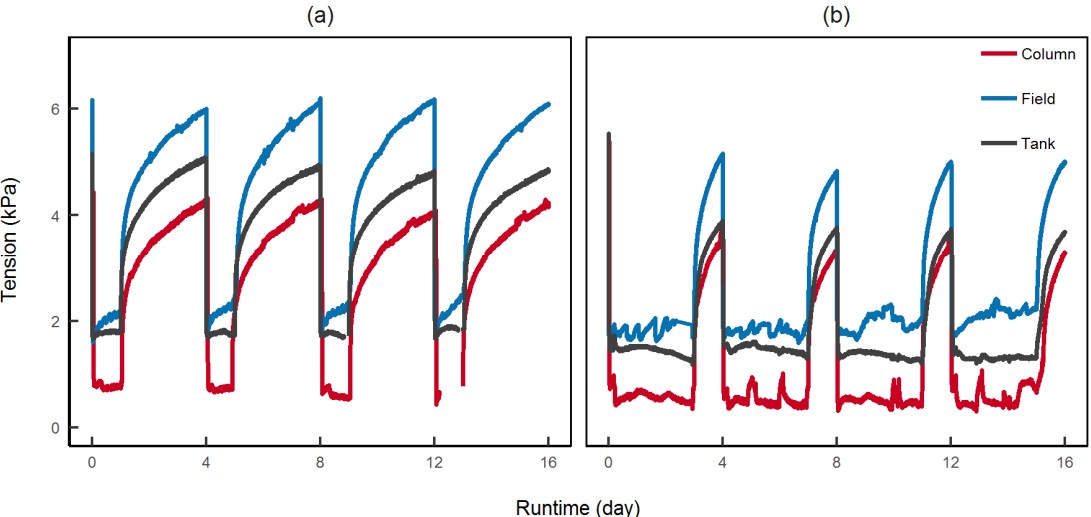

**Figure 6.** Measured matric potential at point 2.2 in all experimental setups during (**a**) scenario 1 (wet/dry ratio: 24 h/72 h) and (**b**) scenario 4 (wet/dry: 72 h/24 h).

This comparison of tension measurements shows that all three systems can be used to reproduce water flow in the unsaturated zone during MAR experiments, as all were able to depict temporal behavior of wetting and drying patterns. Due to their dimensionality, tension values in the column are always below those of the other systems. Thus, minimum and maximum tension values retrieved from column studies are not reliable for representation of 3D field systems.

### 3.3. Oxygen Consumption

For the assessment of oxygen consumption, scenario 3 (continuous infiltration) and scenario 4 (3 days infiltration, 1 day drying) were compared (Figure 7). The results show the effect of including drying cycles on the oxygen consumption and recovery.

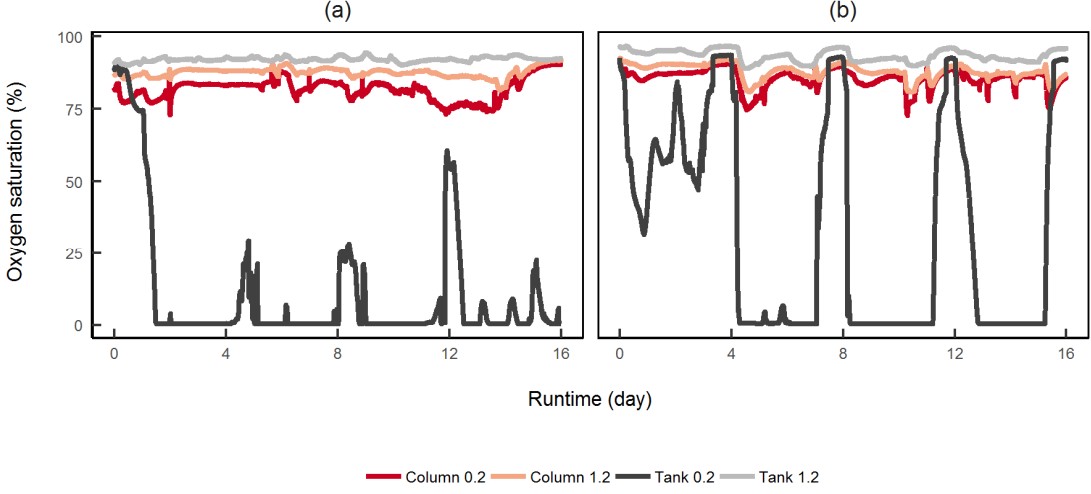

**Figure 7.** Oxygen consumption in point 0.2 and 1.2 during (**a**) scenario 3 (continuous infiltration) and (**b**) scenario 4 (wet/dry ration: 72 h/24 h) for column and tank.

Scenario 3 with continuous infiltration shows high values of oxygen saturation in 28 cm depths in both the tank and column. Oxygen values in the highest column layer (16 cm deep) are slightly lower but continue to be relatively constant throughout the scenario. Only oxygen in the upper layer of the tank is completely depleted throughout the course of infiltration. This indicates that in the upper layer of the tank, oxygen was consumed due to high biological activity. Lateral water flow in the case of ponding and the lower water content in the tank were the reasons for higher biological growth and activity combined with higher consumption of oxygen in phases with water infiltration. Poor conditions for biological growth and activity (continuous ponding respectively to high water content) lead to low consumption of oxygen in the upper layer of the column during phases with water infiltration.

In the case of the scenario with dry phases (scenario 4), the conditions for the growth and activity of the bacteria in the column and tank were better over the runtime of the complete scenario due to the sufficient supply of the bacteria with oxygen as well as more optimal soil moisture contents guaranteed by the existing, dry phases. Again, strong consumption of oxygen could only be detected in the upper layer of the tank, while all other measuring points show only slight changes in oxygen saturation. Scenario 4 depicts clearly that during each drying phase (at day 3, 7, 11 and 15) the oxygen levels rose to 100% and can be fully recovered. Low oxygen consumption during the water infiltration was again measured in the column. Biological activity in phases with water infiltration was not that high due to the insufficient delivery of oxygen over the ponded soil surface and the closed side walls.

## 4. Conclusions

The unique feature of the experimental setup presented in this study is the possibility to compare managed aquifer recharge experiments at different levels of dimension and scale. The focus on hydraulic comparison with assessment of the infiltration capacity reduction, as well as comparison of water flow and oxygen dynamics in all three systems, allows for an analysis of which restrictions and limitations apply to the particular systems. Results of infiltration experiments in all physical models indicate that 3D experiments (field, laboratory tank) have the advantage to reproduce water flow processes and oxygen consumption in unsaturated soil during operation of MAR closer to natural

conditions. This is advantageous for a precise analysis of infiltration processes in general as well as infiltration processes during operation of MAR. Water flow velocity, water saturation and oxygen consumption are often overestimated in 1D-column experiments due to sidewall effects and no lateral flow. Nevertheless, the less costly and less time consuming column experiments are a good opportunity for initial assessment of processes taking place in the unsaturated soil during operation of MAR, such as clogging development and biodegradation processes. In the case of investigating climate-relevant processes, such as clogging development, performance of field experiments is recommended due to the strong influence of site-specific climate on these processes.

**Author Contributions:** Conceptualization, T.F., F.B. and J.S.; Catalin Stefan Investigation, T.F. and F.B.; Methodology, T.F. and F.B.; Validation, T.F., F.B. and J.S.; Writing—original draft, Thomas Fichtner; Writing—review & editing, T.F., F.B., J.S. and C.S.

**Funding:** This work was supported by the German Federal Ministry of Education and Research (BMBF), grant no. 01LN1311A (Junior Research Group "INOWAS").

**Conflicts of Interest:** The authors declare no conflict of interest. The funders had no role in the design of the study; in the collection, analyses, or interpretation of data; in the writing of the manuscript, or in the decision to publish the results.

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
