# Peer review of "Assessing Managed Aquifer Recharge Processes under Three Physical Model Concepts"

_water, doi:10.3390/w11010107_

Round 1

Reviewer 1 Report

Dear author

The main concern for your paper is the lack of discussion. This makes the paper just a mere report. I suggest to improve the paper via the improvements of the discussion as this is a key part that need to me updated

And more

I found your paper of interest and I wish to comment some minor issues I think can make your paper more relevant and of highest impact in the scientific community

your figures should be improved. Use internal axis tick marks (tradition in science), move the legends inside the box of the graph, remove titles (you have the figure caption) or move inside the box of the graph

Figure 3 use color

Figure 2 use color

There is a need to mention measurement methods to show there is a great interest in the scientific community related to the topic of infiltration... your paper is an applied approach too but this information is relevant for the researcher. See here Di Prima, S., Concialdi, P., Lassabatere, L., Angulo-Jaramillo, R., Pirastru, M., Cerda, A., & Keesstra, S. (2018). Laboratory testing of Beerkan infiltration experiments for assessing the role of soil sealing on water infiltration. Catena167, 373-384. as a Beerkan approach example. Or the use of traditional ring infiltrometer Di Prima, S., Rodrigo-Comino, J., Novara, A., Iovino, M., Pirastru, M., Keesstra, S., & CerdĂ , A. (2018). Soil physical quality of citrus orchards under tillage, herbicide, and organic managements. Pedosphere28(3), 463-477.Di Prima, S., Lassabatere, L., Rodrigo-Comino, J., Marrosu, R., Pulido, M., Angulo-Jaramillo, R., ... & Pirastru, M. (2018). Comparing transient and steady-state analysis of single-ring infiltrometer data for an abandoned field affected by fire in Eastern Spain. Water10(4), 514. Or the use of the minidisk infiltrometer Alagna, V., Di Prima, S., Rodrigo-Comino, J., Iovino, M., Pirastru, M., Keesstra, S. D., ... & CerdĂ , A. (2017). The impact of the age of vines on soil hydraulic conductivity in vineyards in eastern Spain. Water10(1), 14

Your methods are three but to mention other methods will help to improve your paper and achieve the dissemination your research needs

I found also that you paper in the first sentence is related to the issue that climate change and land use are affecting the water recharge, but this is not demonstrated in your paper. Please, make an update of this topic and mention examples...and also the key issue of quality and quantity of water resources is relevant.. see here some examples around the world Narany, T. S., Aris, A. Z., Sefie, A., & Keesstra, S. (2017). Detecting and predicting the impact of land use changes on groundwater quality, a case study in Northern Kelantan, Malaysia. Science of the Total Environment599, 844-853.

Your conclusion section is too long.  I suggest to shorten.... we need a short clear and concise conclusion section

Author Response

Dear author

R: The main concern for your paper is the lack of discussion. This makes the paper just a mere report. I suggest to improve the paper via the improvements of the discussion as this is a key part that need to me updated

Answer: We extended the discussion in several points, specifically regarding the applicability of the experiments.

And more

R: I found your paper of interest and I wish to comment some minor issues I think can make your paper more relevant and of highest impact in the scientific community

R: your figures should be improved. Use internal axis tick marks (tradition in science), move the legends inside the box of the graph, remove titles (you have the figure caption) or move inside the box of the graph

Answer: The figures have been adjusted according to your suggestions. Only for figure 7 it was not possible to fit the legend within the box, thus, we kept it below.

R: Figure 3 use color

Answer: The figures have been adjusted according to your suggestions.

R: Figure 2 use color

Answer: The figures have been adjusted according to your suggestions

R: There is a need to mention measurement methods to show there is a great interest in the scientific community related to the topic of infiltration... your paper is an applied approach too but this information is relevant for the researcher. See here Di Prima, S., Concialdi, P., Lassabatere, L., Angulo-Jaramillo, R., Pirastru, M., Cerda, A., & Keesstra, S. (2018). Laboratory testing of Beerkan infiltration experiments for assessing the role of soil sealing on water infiltration. Catena, 167, 373-384. as a Beerkan approach example. Or the use of traditional ring infiltrometer Di Prima, S., Rodrigo-Comino, J., Novara, A., Iovino, M., Pirastru, M., Keesstra, S., & CerdĂ , A. (2018). Soil physical quality of citrus orchards under tillage, herbicide, and organic managements. Pedosphere, 28(3), 463-477.Di Prima, S., Lassabatere, L., Rodrigo-Comino, J., Marrosu, R., Pulido, M., Angulo-Jaramillo, R., ... & Pirastru, M. (2018). Comparing transient and steady-state analysis of single-ring infiltrometer data for an abandoned field affected by fire in Eastern Spain. Water, 10(4), 514. Or the use of the minidisk infiltrometer Alagna, V., Di Prima, S., Rodrigo-Comino, J., Iovino, M., Pirastru, M., Keesstra, S. D., ... & CerdĂ , A. (2017). The impact of the age of vines on soil hydraulic conductivity in vineyards in eastern Spain. Water, 10(1), 14

Your methods are three but to mention other methods will help to improve your paper and achieve the dissemination your research needs

Answer: Further usually used methods for determining hydraulic conductivity in the unsaturated soil zone were added to chapter “2.3. Determination of infiltration capacity reduction”

R: I found also that your paper in the first sentence is related to the issue that climate change and land use are affecting the water recharge, but this is not demonstrated in your paper. Please, make an update of this topic and mention examples ... and also the key issue of quality and quantity of water resources is relevant - see here some examples around the world Narany, T. S., Aris, A. Z., Sefie, A., & Keesstra, S. (2017). Detecting and predicting the impact of land use changes on groundwater quality, a case study in Northern Kelantan, Malaysia. Science of the Total Environment, 599, 844-853.

Answer: Examples for the influence of overexploitation and climate change on groundwater resources and resulting influence of water quality were added 

R: Your conclusion section is too long.  I suggest to shorten.... we need a short clear and concise conclusion section

Answer: Was shortened to make the conclusion more clear and concise

Reviewer 2 Report

A clearly written and easy to understand study on clogging and infiltration rates on MAR systems. some minor comments below:

Page 40 – Actually a revision of many of these MAR schemes shows that the common reason for failure of MAR projects is actually economics:

West, Camilla & Kenway, Steven & Hassall, Maureen & Yuan, Zhiguo. (2016). Why do residential recycled water schemes fail? A comprehensive review of risk factors and impact on objectives. Water Research. 102. 10.1016/j.watres.2016.06.044.

Line 87 – flow rates in column studies are also affected very much effected by temperature – viscosity and the influence on hydraulic permeability, K.

Line 98 – give the temperatures here in brackets for the reader.

Somewhere in the manuscript it wopuld be good to also discuss the various batch studies that can be performed and what they say about clogging -  clay dispersion and metals release etc...?

Line 207 =- shape of the clogging curve also can give information on the dominant mechanism.

Rinck-Pfeiffer, Stéphanie & Ragusa, Santo & Sztajnbok, Pascale & Vandevelde, Thierry. (2000). Interrelationships between biological, chemical, and physical processes as an analog to clogging in aquifer storage and recovery (ASR) wells. Water Research. 34. 2110-2118. 10.1016/S0043-1354(99)00356-5.

Figure 4 – suggest change the scale to m3/m2 expressed simply as m? This would be SI units and preferred for the journal I assume and is the standard way to express filters in water treatment for example.

Line 227 – I’d say there are competing forces of bioclogging and higher temperatures and less infiltration at lower temperatures due to viscosity changes. I don’t feel that the conclusion is justified that higher temperatures don’t effect clogging due to the confounding nature of the experiment. If a temperature probe had been used and hydraulic permeability plotted then this could be answered.

Line 259 - Figure 6 – previous figures results placed field results between column and tank but not this one. Why the difference here? An explanation is required that flows down to the conclusion.

Line 307 – I don’t agree that the 3D setup performs much better to the field trial for the results given. It’s different to the column but as no measure of why it is superior is given then no such assumption can be made. If anything it seems from the experiments – the field seems to lie between the extremes of the column and tank for the flow rates (Figures 4 and 5) but not for matric potential. My own conclusion from reading the work is that a 3D model is probably not worth the effort as it doen's perform that much better than a column. (unless you give a measure of the performance it could be interpreted either way).

Author Response

R: Page 40 – Actually a revision of many of these MAR schemes shows that the common reason for failure of MAR projects is actually economics:

West, Camilla & Kenway, Steven & Hassall, Maureen & Yuan, Zhiguo. (2016). Why do residential recycled water schemes fail? A comprehensive review of risk factors and impact on objectives. Water Research. 102. 10.1016/j.watres.2016.06.044.

Answer: Economics as a shutdown reason for MAR schemes was included

R: Line 87 – flow rates in column studies are also affected very much effected by temperature – viscosity and the influence on hydraulic permeability,

Answer: It was mentioned from line 146 that the influence of temperature on water viscosity and therefore on hydraulic permeability can lead to over- or underestimation of clogging strength. 

R: Line 98 – give the temperatures here in brackets for the reader.

Answer: Temperatures were added

R: Somewhere in the manuscript it would be good to also discuss the various batch studies that can be performed and what they say about clogging - clay dispersion and metals release etc...?

Answer: The topic clay dispersion and metals release as processes influencing operation of MAR was included in the chapter “1 Introduction”

R: Line 207 =- shape of the clogging curve also can give information on the dominant mechanism.

Rinck-Pfeiffer, Stéphanie & Ragusa, Santo & Sztajnbok, Pascale & Vandevelde, Thierry. (2000). Interrelationships between biological, chemical, and physical processes as an analog to clogging in aquifer storage and recovery (ASR) wells. Water Research. 34. 2110-2118. 10.1016/S0043-1354(99)00356-5.

Answer: We added a discussion about the dominant clogging mechanism in the text passage describing figure 4.

R: Figure 4 – suggest change the scale to m3/m2 expressed simply as m? This would be SI units and preferred for the journal I assume and is the standard way to express filters in water treatment for example.

Answer:  The unit for specific volume has been changed to mÂł/m².

R. Line 227 – I’d say there are competing forces of bioclogging and higher temperatures and less infiltration at lower temperatures due to viscosity changes. I don’t feel that the conclusion is justified that higher temperatures don’t effect clogging due to the confounding nature of the experiment. If a temperature probe had been used and hydraulic permeability plotted then this could be answered.

Answer: The topic of competing forces of bioclogging and higher temperatures as well as less infiltration at lower temperatures due to viscosity changes is now discussed in chapter “1 Introduction” and “3.1 Infiltration capacity reduction”

R: Line 259 - Figure 6 – previous figures results placed field results between column and tank but not this one. Why the difference here? An explanation is required that flows down to the conclusion.

Answer: The reason why field measurements sometimes lie between those of the laboratory studies and for figure 6 above those, lies within the depicted parameters themselves. Figure 4 and 5 depict flow rates or decrease of velocities. Figure 6 depicts tension or matric potential measurements. Both parameters are related but can still behave differently in the various systems. It must be further added that e.g. for figure 4 the measurements of the soil column were not completed over the entire time frame due to overflow/clogging. Looking at figure 4a it can be assumed that if further measurements were available the decrease of velocity would have dropped even further, below those values of the field. Thus, your general assumption of the field laying in between column and tank measurements cannot be stated that clearly.

R: Line 307 – I don’t agree that the 3D setup performs much better to the field trial for the results given. It’s different to the column but as no measure of why it is superior is given then no such assumption can be made. If anything it seems from the experiments – the field seems to lie between the extremes of the column and tank for the flow rates (Figures 4 and 5) but not for matric potential. My own conclusion from reading the work is that a 3D model is probably not worth the effort as it doen's perform that much better than a column. (unless you give a measure of the performance it could be interpreted either way).

Answer: We shortened the conclusion and changed the wording so that the differences in obtained results from 1D and 3D experiments becomes more apparent. We also avoided to call one system the “better option” in general even though advantages and disadvantages of the systems must be mentioned.

Round 2

Reviewer 1 Report

Dear author

Right now your paper reads as a report. There is a need that you will jump to show your findings as a key contribution to science and for this the discussion section should be improved with comments to the implications of your findings and comments to other researches

Sincerely

Reviewer 2 Report

No further comments